# Learning Deep-Latent Hierarchies by Stacking Wasserstein Autoencoders

## Abstract

Probabilistic models with hierarchical-latent-variable structures provide state-of-the-art results amongst non-autoregressive, unsupervised density-based models. However, the most common approach to training such models based on Variational Autoencoders often fails to leverage deep-latent hierarchies; successful approaches require complex inference and optimisation schemes. Optimal Transport is an alternative, non-likelihood-based framework for training generative models with appealing theoretical properties, in principle allowing easier training convergence between distributions. In this work we propose a novel approach to training models with deep-latent hierarchies based on Optimal Transport, without the need for highly bespoke models and inference networks. We show that our method enables the generative model to fully leverage its deep-latent hierarchy, and that in-so-doing, it is more effective than the original Wasserstein Autoencoder with Maximum Mean Discrepancy divergence.

## 1 Introduction

Probabilistic latent-variable modelling is widely applicable in machine learning as a method for discovering structure directly from large, unlabelled datasets. Variational Autoencoders (VAEs) (Kingma & Welling, 2014; Rezende et al., 2014) have proven to be effective for training generative models parametrised by powerful neural networks, by mapping the data into a low-dimensional embedding space. While allowing the training of expressive models, VAEs often fail when using deeper hierarchies with several stochastic latent layers.

In fact, many of the most successful probabilistic latent-variable models use only one stochastic latent layer. Auto-regressive models (Larochelle & Murray, 2011; Van Den Oord et al., 2016b;a; Salimans et al., 2017), for example, produce state-of-the-art samples and likelihood scores. However, auto-regressive models suffer from poor scalability to high-dimensional data. Flow-based models (Rezende & Mohamed, 2015; Kingma et al., 2016; Dinh et al., 2016; Kingma & Dhariwal, 2018) are another class of generative models providing competitive sample quality and are able to scale to higher-dimensional data, but they still lag behind auto-regressive models in terms of likelihood score.

Deep-latent-variable models are highly expressive models that aim to capture the structure of data in a hierarchical manner, and could thus potentially compete with auto-regressive models for state-of-the-art performance. However, they remain hard to train. Many explanations have been proposed for this, from the use of dissimilarity measure directly in pixel space (Larsen et al., 2016) resulting in poor sample quality, to the lack of efficient representation in the latent (Zhao et al., 2017), to simply the poor expressiveness of the models used (Zha et al., 2017; Maaløe et al., 2019).

Recent works have tried to overcome the latter problem introducing latent skip connections in the generative model and inference network (Bachman, 2016), sharing generative model and inference network parameters (Sønderby et al., 2016), as well as bidirectional inference network (Maaløe et al., 2019). Maaløe et al. (2019) managed to train very deep-hierarchical-latent models achieving near state-of-the-art sample generations. However, in order to leverage their latent hierarchy (working in the VAE framework), they needed both complex, tailored inference networks, and deterministic skip connections in the generative model.

Optimal Transport (OT) (Villani, 2008) is a mathematical framework to compute distances between distributions. Recently, it has been used as a training method for generative models (Genevay et al.,

2018; Bousquet et al., 2017). Tolstikhin et al. (2018) introduced Wasserstein Autoencoders (WAEs), where as with VAEs, an encoding distribution maps the data into a latent space, aiming to learn a low-dimensional representation of the data. WAE is a non-likelihood based framework with appealing topological properties (Arjovsky et al., 2017; Bousquet et al., 2017), in theory allowing for easier training convergence between distributions. Gaujac et al. (2018) trained a two-latent-layer generative model using WAE, showing promising results in the capacity of the WAE framework to leverage a latent hierarchy in generative modelling.

Following these early successes, we propose to train deep-hierarchical-latent models using the WAE framework, without the need for complex dependency paths in both the generative model and inference network. As in the works of Sønderby et al. (2016) and Maaløe et al. (2019), we believe that a deep-latent hierarchy offers the potential to improve generative models, if they could be trained properly. In order to leverage the deep-latent hierarchy, we derive a novel objective function by stacking WAEs, introducing an inference network at each level, and encoding the information up to the deepest layer in a bottom-up way. For convenience, we refer to our method as STACKEDWAE.

Our contributions are two-fold: first, we introduce STACKEDWAE, a novel objective function based on OT, designed specifically for generative modelling with latent hierarchies, and show that it is able to fully leverage its hierarchy of latents. Second, we show that STACKEDWAE performs significantly better when training hierarchical-latent models than the original WAE framework regularising the latent with the Maximum Mean Discrepancy (MMD) divergence (Gretton et al., 2012).

## 2 STACKED WAE

STACKEDWAEs are probabilistic-latent-variable models with a deep hierarchy of latents. They can be minimalistically simple in their inference and generative models, but are trained using OT in a novel way. We start by defining the probabilistic models considered in this work, then introduce OT, and finally discuss how to train probabilistic models with deep-latent hierarchies using OT, the method that we refer to as STACKEDWAE.

Throughout this paper, we use calligraphic letters (e.g. $\mathcal{X}$) for sets, capital letters (e.g. $X$) for random variables, and lower case letters (e.g. $x$) for their values. We denote probability distributions with capital letters (e.g. $P(X)$) and their densities with lower case letters (e.g. $p(x)$).

### 2.1 GENERATIVE MODELS WITH DEEP HIERARCHIES OF LATENT VARIABLES

We will consider deep-generative models with Markovian hierarchies in their latent variables. Namely, where each latent variable depends exclusively on the previous one. Denoting by $P_\Theta$ the parametric model with $N$ latent layers, where $\Theta = \{\theta_1, \dots, \theta_N\}$, we have:

$$p_\Theta(x) = \int_{\mathcal{Z}_1 \dots \mathcal{Z}_N} p_{\theta_1}(x|z_1) \, p_{\theta_2}(z_1|z_2) \dots p_{\theta_N}(z_{N-1}|z_N) \, p(z_N) \, dz_1 \dots dz_N \tag{1}$$

where the data is $X \in \mathcal{X}$ and the latent variables are $Z_n \in \mathcal{Z}_n$ and we chose $p(z_N) = \mathcal{N}(z_N; 0_{\mathcal{Z}_N}, \mathcal{I}_{\mathcal{Z}_N})$. The corresponding graphical model for $N = 3$ is given Figure 1a

We will be using variational inference through the WAE framework of Tolstikhin et al. (2018), introducing variational distributions, $q_\Phi(z_1, \dots, z_N|x)$, to facilitate the evaluation of the integral in Eq. (1), analogous to VAEs (Kingma & Welling, 2014; Rezende et al., 2014). It will be shown in Section 2.3 that without loss of generality, $q_\Phi(z_1, \dots, z_N|x)$ can have a Markovian latent hierarchy when following the STACKEDWAE approach. That is,

$$q_\Phi(z_1, \dots, z_N|x) \overset{\substack{\text{w.l.o.g. with} \\ \text{STACKEDWAE}}}{=} q_{\phi_1}(z_1|x) \, q_{\phi_2}(z_2|z_1) \dots q_{\phi_N}(z_N|z_{N-1}) \tag{2}$$

where each $q_{\phi_N}(z_i|z_{i-1})$ is introduced iteratively by stacking WAEs at each latent layer. The corresponding graphical model for $N = 3$ is given Figure 1b

We focus on this simple Markovian latent-variable structure for the generative model as a proof point for STACKEDWAE. This simple modelling setup is famously difficult to train, as is discussed extensively in the VAE framework (see for example Burda et al. (2015); Sønderby et al. (2016); Zhao et al. (2017)). The difficulty in training such models comes from the Markovian latent structure

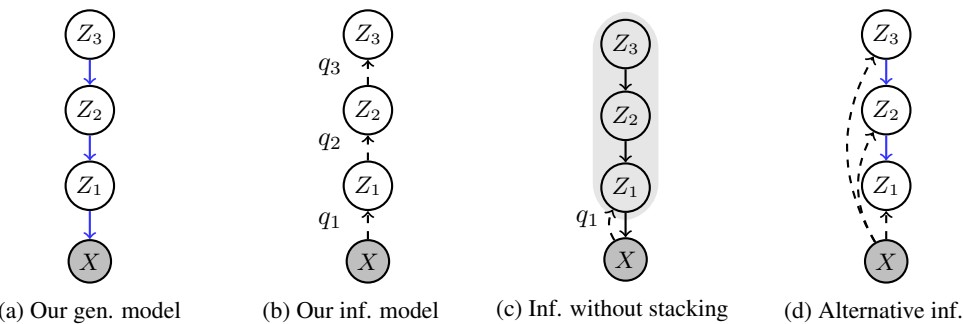

(a) Our gen. model      (b) Our inf. model      (c) Inf. without stacking      (d) Alternative inf.

Figure 1: (a) Generative model (blue lines represent generative model parameters). (b) Inference model used in STACKEDWAE. (c) Standard WAE; the generative model has only one latent with prior $p(z_1) = \int p(z_1|z_2)p(z_2|z_3)p(z_3)dz_1 dz_2 dz_3$. (d) Inference model with skips connections and parameter sharing with the generative model, as in Sønderby et al. (2016).

of the generative model; in particular, the difficulty of learning structure in the deeper latents. This is because, to generate samples $x \sim p_{\Theta}(x)$, only the joint $p_{\Theta}(x, z_1)$ is needed as $p_{\Theta}(x) = \int_{\mathcal{Z}_1} p_{\theta_1}(x|z_1)p_{\Theta}(z_1)dz_1$. Learning a smooth structure in each latent layer is not a strict requirement for learning a good generative model, however, it is sought after if the latent is to be used downstream or interpreted. We find empirically (see Section 3.1) that a better generative model is also achieved when the latent hierarchy is well learnt all the way down.

Sønderby et al. (2016) sought to overcome the difficulty of learning a deep-latent hierarchy by using deterministic bottom-up inference, followed by top-down inference that shared parameters with the generative model. With additional optimisation tricks (e.g. KL annealing), their deeper-latent distributions would still go unused for sufficiently deep-latent hierarchies (discussed in Maaløe et al. (2019)). In order to get deeper hierarchies of latents, Maaløe et al. (2019) introduced additional deterministic connections in the generative model as well as bidirectional inference network to facilitate the deep information flow needed to ensure the usage of the deeper-latent layers.

While the approach in Maaløe et al. (2019) is well motivated and achieves excellent results, we choose the OT framework for training deep-latent models due to its topological properties (see Arjovsky et al. (2017)). Still, the standard WAE encounters the same difficulties as the VAE in learning deep-latent hierarchies. We thus modify the original WAE objective, effectively stacking WAEs, to improve the learning of both the generative model and the inference distribution throughout the latent hierarchy.

## 2.2 WASSERSTEIN AUTOENCODERS

The Kantorovich formulation of the OT between the true-but-unknown data distribution $P_D$ and the model distribution $P_{\Theta}$, for a given cost function $c$, is defined by:

$$\mathrm{OT}_c(P_D, P_{\Theta}) = \inf_{\Gamma \in \mathcal{P}(P_D, P_{\Theta})} \int_{\mathcal{X} \times \mathcal{X}} c(x, \tilde{x}) \, d\Gamma(x, \tilde{x}) \tag{3}$$

where $\mathcal{P}(P_D, P_{\Theta})$ is the space of all couplings of $P_D$ and $P_{\Theta}$; namely, the space of joint distributions $\Gamma$ on $\mathcal{X} \times \mathcal{X}$ whose densities $\gamma$ have marginals $p_D$ and $p_{\Theta}$:

$$\mathcal{P}(P_D, P_{\Theta}) = \left\{ \Gamma \ \middle| \ \int_{\mathcal{X}} \gamma(x, \tilde{x}) \, d\tilde{x} = p_D(x) \ \text{ and } \ \int_{\mathcal{X}} \gamma(x, \tilde{x}) \, dx = p_{\Theta}(\tilde{x}) \right\} \tag{4}$$

In the WAE framework introduced by Tolstikhin et al. (2018), the space of couplings is constrained to joint distributions $\Gamma$, with density $\gamma$, of the form:

$$\gamma(x, \tilde{x}) = \int_{\mathcal{Z}_1 \dots \mathcal{Z}_N} p_{\Theta}(\tilde{x}|z_1, \dots, z_N) \, q_{\phi_1}(z_1, \dots, z_N|x) \, p_D(x) \, dz_1 \dots dz_N \tag{5}$$

where $Q_{\phi_1}(Z_1, \dots, Z_N|X)$ plays the same role as the variational distribution in variational inference.

Marginalising over $\tilde{x}$ in Eq. (5) automatically gives $p_D(x)$, however the second marginal constraint in Eq. (4) (that over $x$ giving $p_\Theta$) is not guaranteed. Due to the Markovian structure of the generative model, a sufficient condition for satisfying the second marginal constraint is (see Appendix A.1):

$$\int_{\mathcal{X}} q_{\phi_1}(z_1|x) p_D(x) dx = \int_{\mathcal{Z}_2 \dots \mathcal{Z}_N} p_{\theta_2}(z_1|z_2) \dots p_{\theta_N}(z_{N-1}|z_N) p(z_N) dz_2 \dots dz_N \overset{\text{def}}{=} p_{\Theta_{2:N}}(z_1) \tag{6}$$

Finally, to get the WAE objective of Tolstikhin et al. (2018), the marginal constraint of Eq. (6) is relaxed using a Lagrange multiplier (see Appendix A.2):

$$\widehat{W}_c(P_D, P_\Theta) = \inf_{Q_{\phi_1}(Z_1|x)} \left[ \lambda_1 \mathcal{D}_1 \left( \int_{\mathcal{X}} Q_{\phi_1}(Z_1|x) p_D(x) dx, \ P_{\Theta_{2:N}}(Z_1) \right) \right. \tag{7}$$

$$\left. + \int_{\mathcal{X} \times \mathcal{X}} \int_{\mathcal{Z}_1} c(x, \tilde{x}) p_{\theta_1}(\tilde{x}|z_1) q_{\phi_1}(z_1|x) p_D(x) dx d\tilde{x} dz_1 \right]$$

where $\mathcal{D}_1$ is any divergence function and $\lambda_1$ a relaxation parameter. Note that the inf is taken only over $q_{\phi_1}(z_1|x)$ instead of the full $q_{\phi_1}(z_1, \dots, z_N|x)$ because the expression does not depend on $z_{>1}$.

While Eq. (7) is in-principle tractable (e.g. for Gaussian $q_{\phi_1}(z_1|x)$ and sample-based divergence function such as MMD), it only depends on the first latent $Z_1$. Thus it will learn only a good approximation for $p_\Theta(x, z_1) = p_{\theta_1}(x|z_1)p(z_1)$, rather than a full hierarchy with each latent living on a smooth manifold. We show empirically in Section 3.1 that Eq. (7) is indeed insufficient.

## 2.3 STACKING WAEs FOR DEEP-GENERATIVE-LATENT VARIABLE MODELLING

In theory (e.g. $\lambda_1 \to \infty$), Eq. (7) does not depend on the choice of divergence $\mathcal{D}_1$. However, given the set of approximations used, a divergence that takes into account the smoothness of the full stack of latents will likely help the optimisation. We now show that by using the Wasserstein distance itself for $\mathcal{D}_1$, we can derive an objective that naturally pairs up inference and generation at every level in the deep-latent hierarchy. After all, the divergence in Eq. (7) is between an aggregate distribution:

$$Q_1^{\text{agg}}(Z_1) \overset{\text{def}}{=} \int_{\mathcal{X}} Q_{\phi_1}(Z_1|x) p_D(x) dx \tag{8}$$

from which we can only access samples, and an analytically-known distribution $P_{\Theta_{2:N}}(Z_1)$, which is analogous to where we started with the Wasserstein distance between $P_D$ and the full $P_\Theta$.

Specifically, we choose $\mathcal{D}_1$ in Eq. (7) to be the relaxed Wasserstein distance $\widehat{W}_{c_1}$, which following the same arguments as before, requires the introduction of a new variational distribution $Q_{\phi_2}(Z_2|Z_1)$:

$$\mathcal{D}_1 \left( Q_1^{\text{agg}}(Z_1), \ P_{\Theta_{2:N}}(Z_1) \right) = \inf_{Q_{\phi_2}(Z_2|z_1)} \left[ \lambda_2 \mathcal{D}_2 \left( Q_2^{\text{agg}}(Z_2), \ P_{\Theta_{3:N}}(Z_2) \right) \right. \tag{9}$$

$$\left. + \int_{\mathcal{Z}_1 \times \mathcal{Z}_1} \int_{\mathcal{Z}_2} c_1(z_1, \tilde{z}_1) p_{\theta_2}(\tilde{z}_1|z_2) q_{\phi_2}(z_2|z_1) q_1^{\text{agg}}(z_1) dz_1 d\tilde{z}_1 dz_2 \right]$$

where the notation in Eq. (6) has been re-used for the prior $P_{\Theta_{3:N}}(Z_2)$, and similarly for the aggregated posterior, $Q_2^{\text{agg}}(Z_2) \overset{\text{def}}{=} \int_{\mathcal{Z}_1} Q_{\phi_2}(Z_2|z_1) q_1^{\text{agg}}(z_1)$, as in Eq. (8). Just as before, $Q_{\phi_2}(Z_2|Z_1)$ does not need to provide a distribution over the $z_{>2}$ without loss of generality.

The divergence $\mathcal{D}_1$ that arose in Eq. (7) between two distributions over $Z_1$ is thus mapped onto the latent at the next level in the latent hierarchy, $Z_2$, via Eq. (9). This process can be repeated by using $\widehat{W}_{c_2}$ again for $\mathcal{D}_2$ in Eq. (9) to get an expression that maps to $Z_3$, requiring the introduction of another variational distribution $Q_{\phi_3}(Z_3|Z_2)$. Repeating this process until we get to the final layer of the hierarchical-latent-variable model gives the STACKEDWAE objective:

$$W_{\text{STACKEDWAE}}(P_D, P_\Theta) = \inf_{Q_\Phi(Z_1, \dots, Z_N|x)} \left( \left[ \prod_{i=1}^{N} \lambda_i \right] \mathcal{D}_N \left( Q_N^{\text{agg}}(Z_N), \ P(Z_N) \right) \right. \tag{10}$$

$$\left. + \sum_{n=0}^{N-1} \left[ \prod_{i=1}^{n} \lambda_i \right] \int_{\mathcal{Z}_n \times \mathcal{Z}_n} \int_{\mathcal{Z}_{n+1}} c_n(z_n, \tilde{z}_n) p_{\theta_{n+1}}(\tilde{z}_n|z_{n+1}) q_{\phi_{n+1}}(z_{n+1}|z_n) q_n^{\text{agg}}(z_n) dz_n d\tilde{z}_n dz_{n+1} \right)$$

where we denote $(\mathcal{Z}_0, Z_0, z_0) = (\mathcal{X}, X, x)$ and we define the empty product $\prod_{i=1}^{0} \lambda_i \overset{\text{def}}{=} 1$. Each $p_{\theta_n}$ is the $n^{\text{th}}$ layer of the generative model given in Eq. (1). The $q_{\phi_n}$'s are the inference models introduced each time a WAE is "stacked", which combine to make the overall STACKEDWAE Markovian inference model given in Eq. (2), and the aggregated posterior distributions are defined as

$$Q_0^{\text{agg}} \overset{\text{def}}{=} P_D \qquad \text{and} \qquad Q_n^{\text{agg}}(Z_n) \overset{\text{def}}{=} \int_{\mathcal{Z}_{n-1}} Q_{\phi_n}(Z_n|z_{n-1}) \, q_{n-1}^{\text{agg}}(z_{n-1}) \, dz_{n-1} \qquad (11)$$

Note that the STACKEDWAE objective function in Eq. (10) still requires the specification of a divergence at the highest latent layer $\mathcal{D}_N$, which we simply take to be the MMD as originally proposed by Tolstikhin et al. (2018). Other choices can be made, as in Patrini et al. (2018), who choose a Wasserstein distance computed using the Sinkhorn algorithm (Cuturi, 2013). While Patrini et al. (2018) provide a theoretical justification for the minimisation of a Wasserstein distance in the prior space, we found that it did not result in significant improvement and comes at an extra efficiency cost. Similarly, one could choose different cost functions $c_n$ at each layer; for simplicity we take all cost functions to be the squared Euclidean distance in their respective spaces.

## 3    EXPERIMENTS

We now show how the STACKEDWAE approach of Section 2 can be used to train deep-latent hierarchies without customizing the generative or inference models (e.g. skip connections, parameter sharing) . We also show through explicit comparison that the STACKEDWAE approach performs significantly better than the standard WAE when training deep-hierarchical-latent models.

### 3.1    MNIST

#### EXPERIMENTAL SETUP

We trained a deep-hierarchical-latent variable model with $N = 5$ latent layers on raw (non-binarised) MNIST (LeCun & Cortes, 2010). The latent layers have dimensions: $d_{\mathcal{Z}_1} = 32$, $d_{\mathcal{Z}_2} = 16$, $d_{\mathcal{Z}_3} = 8$, $d_{\mathcal{Z}_4} = 4$ and $d_{\mathcal{Z}_5} = 2$. We chose Gaussian distributions for both the generative and inference models, except for the bottom layer of the generative model, which we choose to be deterministic, as in Tolstikhin et al. (2018). The mean and covariance matrices used are parametrised by fully connected neural networks (see Appendix B.1 for details).

We choose the squared Euclidean distance cost function: $c_n(z_n, \tilde{z}_n) = \|z_n - \tilde{z}_n\|_{L^2}^2$. Expectations in Eq. (10) are computed analytically where possible, and otherwise with Monte Carlo sampling.

#### LEARNING A DEEP-LATENT HIERARCHY

Our results are shown in Figure 2, with the training curves in Figure 2a, samples from the generative model in Figure 2b, and latent space interpolations in Figure 2c where digits are reconstructed from points in $\mathcal{Z}_5$ taken evenly in a grid varying between $\pm 2$ standard deviations from the origin. STACKEDWAE generates compelling samples and learns a smooth manifold. Note that the choice $d_{\mathcal{Z}_5} = 2$ allows for easy visualisation of the learned manifold, rather than being the optimal dimension for capturing all of the variance in MNIST.

Figure 2c shows that STACKEDWAE manages to use all of its latent layers, capturing most of the covariance structure of the data in the deepest-latent layer, which is something that VAE methods struggle to accomplish (Sønderby et al., 2016). Figure 3a shows the encoded input images through the latent layers, with corresponding digit labels coloured. We see through each layer that STACKEDWAE leverages the full hierarchy in its latents, with structured manifolds learnt at each stochastic layer.

An advantage of deep-latent hierarchies is their capacity to capture information at different levels, augmenting a single-layer latent space. In Figure 3b, input images, shown in the bottom row, are encoded and reconstructed for each latent layer. More specifically, the inputs are encoded up to the latent layer $i$ and reconstructed from the encoded $z_i$ using the generative model $p(x|z_1)p(z_1|z_2)\ldots p(z_{i-1}|z_i)$. Row $i$ in Figure 3b shows the reconstruction obtained from encoding up to layer $i$. We can see that each additional encoding layer moves slightly farther away from copying the input image as it moves

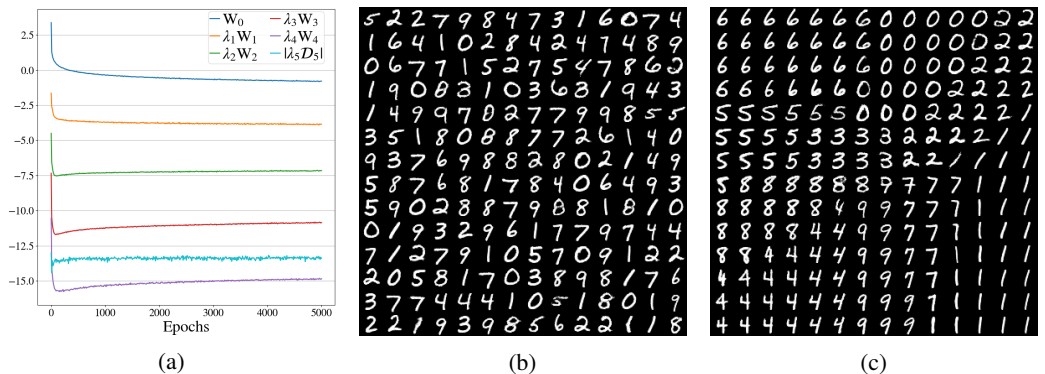

Figure 2: 5-layer STACKEDWAE. (a) Training curves; each term in Eq. (10) is shown throughout training. (b) Model samples. (c) $\mathcal{Z}_5$ latent-space interpolations.

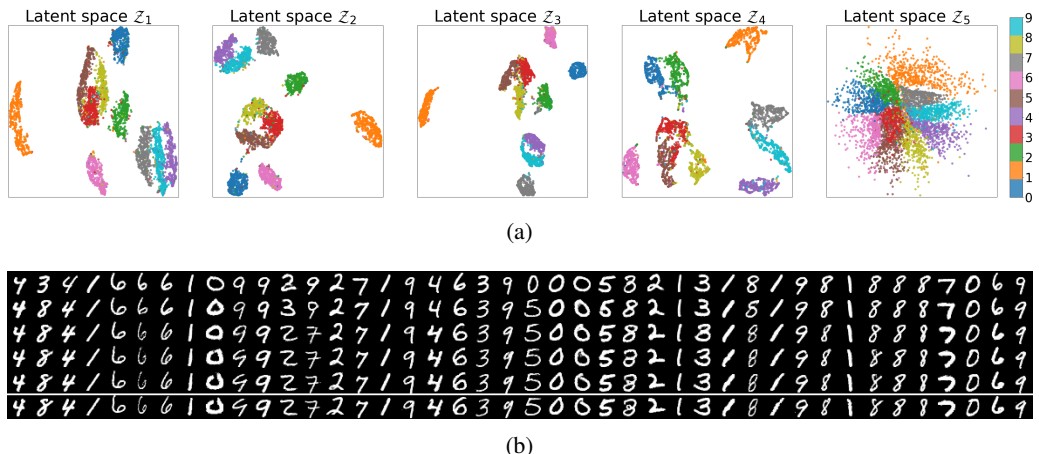

Figure 3: 5-layer STACKEDWAE. (a) Visualisations of latent spaces $\mathcal{Z}_i$. Each colour corresponds to a digit label. $d_{\mathcal{Z}_5} = 2$ can be directly plotted; for higher dimensions we use UMAP (McInnes & Healy, 2018) to plot a two dimensional representation. (b) Reconstructions for different encoding layers. The bottom row is data; the $i^{\text{th}}$ row from the bottom is generated using the latent codes $z_i$ which are from the $i^{\text{th}}$ encoding layer.

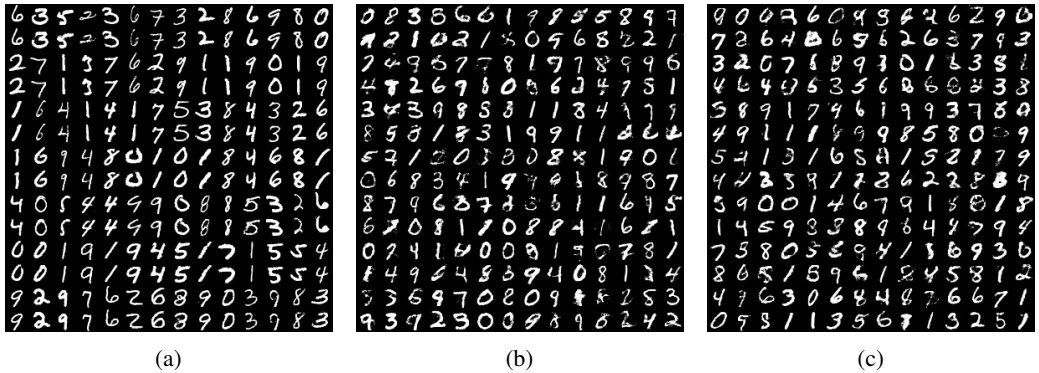

Figure 4: 1-layer implicit-prior WAE. (a) Reconstructions (within pairs of rows, data is above with the corresponding reconstructions below). (b) Model samples. (c) $\mathcal{Z}_5$ latent-space interpolations.

towards fitting the encoding into the 2-dimensional, unit-normal prior distribution. The dimensionality of the deeper-latent layers is a modelling choice which determines how much information is preserved; this can be seen through the loss of information from deeper reconstructions in Figure 3b. Indeed, in each layer, the encoder is asked to map the incoming encodings into a lower-dimensional latent space, filtering the amount of information to keep and pass up to the deeper layer. Thus, if there is a mismatch in the dimensions between the true underlying generative process of the data and the chosen model, the encoder will have to project the encodings into lower-dimensional space, losing information along the way.

ABLATION STUDY: STACKEDWAE VERSUS STANDARD WAE

In this section, we compare STACKEDWAE with the original WAE framework for training generative models with deep-hierarchical latents. In particular, we train a WAE using the objective defined in Eq. (7) and an inference distribution as in Eq. (2); the corresponding graphical model is shown in Figure 1c. We use the same the experimental setup as outlined earlier in this section, and the same parametrised networks.

The results are shown in Figure 4, with reconstructions in Figure 4a, generated samples in Figure 4b, and deepest-latent interpolations in Figure 4c. The latter two should be compared directly with Figure 2b and Figure 2c, respectively, for the STACKEDWAE. The quality of the generated samples from the WAE is poor in comparison with that of the STACKEDWAE.

The lack of smooth interpolations in Figure 4c shows that almost no structure has been captured in the deepest-latent layer. This is likely due to the fact that the standard WAE, with objective given in Eq. (7), is independent of the deeper-latent inference distributions, thus reducing the smoothness in the deeper layers. The relatively accurate reconstructions in Figure 4a indicate that the model managed to capture most of the structure of the data in the first latent layer. This behaviour is similar to that of the Markov HVAE as described in Zhao et al. (2017). They show that, for Markov HVAEs to learn interpretable latents, one needs additional constraints beyond simply maximising the likelihood.

## 3.2 STREET VIEW HOUSE NUMBERS

We now show that STACKEDWAE is able to leverage a deep-latent hierarchy on more real-world datasets, in particular Street View House Numbers (SVHN) (Netzer et al., 2011). We trained a 6-layer STACKEDWAE with inference networks and generative models at each latent layer taken to be Gaussian distributions with mean and covariance functions parametrised by 3-layer ResNet-style (Kaiming et al., 2015) neural networks. The details are given in Appendix B.2.

As before, we choose $c_n(z_n, \tilde{z}_n) = \|z_n - \tilde{z}_n\|_{L^2}^2$, and compute the expectations in Eq. (10) analytically whenever possible, otherwise using Monte Carlo sampling.

For simplicity, our architecture choices constrain the dimensionality on the latent space (details are in Appendix B.2). The latent dimension is given by the size of the output feature map at that layer times the number of these feature maps, leading to latent dimensions of: 16×16×1, 8×8×2, 4×4×4, 4×4×2, 4×4×1 and 2×2×2. These make for relatively high-dimensional latent spaces for the first few latent layers. Rubenstein et al. (2018) observed that when training WAEs with high-dimensional latent space, the variance of the inference network tends to collapse to 0. The authors argue that this might come either from an optimisation issue or from the failing of the divergence used to regularise the latents. Either way, the collapse to deterministic encoders results in poor sample quality as the deterministic encoder is being asked to map the input manifold into a higher-dimensional space than its intrinsic dimension. One solution proposed in Rubenstein et al. (2018) is to include in the objective function Eq. (10) a regulariser that maintains a non-zero variance. As in Rubenstein et al. (2018), we include a log-variance penalty term as given in Eq. (12):

$$\mathcal{L}_{\text{pen}} = \sum_{i=1}^{N} \lambda_i^{\Sigma} \sum_{m=1}^{d_{\mathcal{Z}_i}} |\log \Sigma_i^q[m]| \tag{12}$$

We find that $\lambda_i^{\Sigma} = 10^{-(2+i)}$ for $i = 1, \ldots, 6$ works well in our setting. This choice has been motivated by the fact that the bigger the latent dimension, the more likely it is that $d_{\mathcal{Z}} > d_{\mathcal{I}}$, where $d_{\mathcal{I}}$ and $d_{\mathcal{Z}}$ are the intrinsic dimension of the input and the latent dimension, respectively. Given our

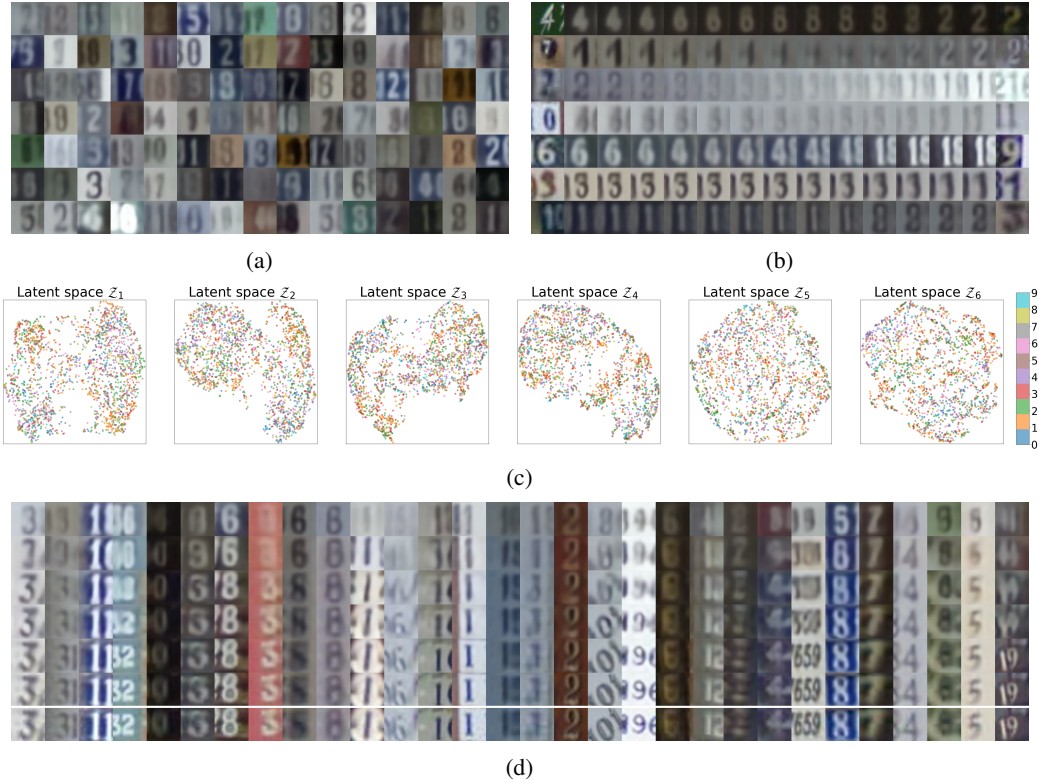

Figure 5: 6-layer STACKEDWAE. (a) Model samples. (b) Points interpolations; the first and last columns are actual data points. (c) Visualisations of latent spaces $\tilde{\mathcal{Z}}_i$, as in Figure 3a. (d) Reconstructions for different encoding layers, as in Figure 3b.

latent-dimension choices (see Table 6b), the first latent layers will suffer more from the collapse to deterministic encoder than the deeper ones with lower dimension.

Our results on SVHN are given Figure 5. Samples from the generative model are shown in Figure 5a. Interpolations across the deepest-latent layer are shown in Figure 5b with the anchor data points in the first and last columns; the interpolations are smooth as expected, though perhaps due to the choice of low-dimensional deepest layers, the reconstructions (second and second-to-last columns) may fail to match perfectly the anchor data points (first and last columns). The encoded latent space is shown in Figure 5c for each latent layer, where the structure of SVHN can be seen; it is likely that the most salient structure in these embeddings represents the background colour, rather than the central digit, but that is to be expected from SVHN. Finally, Figure 5d shows the reconstructions of the data points (along the bottom row) at each latent layer in the hierarchy. Similarly to the results for MNIST, we can see that the deepest latent layer may not be large enough to enable high-fidelity reconstructions. Our intention is to show that the hierarchy of latents can be learnt, which is clearly the case, rather than to model SVHN perfectly, so we do not attempt to tune to the optimal latent dimensionality.

## 4 CONCLUSION

In this work we introduced a novel objective function for training generative models with deep hierarchies of latent variables using Optimal Transport. Our approach recursively applies the Wasserstein distance as the regularisation divergence, allowing the stacking of WAEs for arbitrarily deep-latent hierarchies. We showed that this approach enables the learning of smooth latent distributions even in deep latent hierarchies, which otherwise requires extensive model design and tweaking of the optimisation procedure to train. We also showed that our approach is significantly more effective at learning smooth hierarchical latents than the standard WAE.

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

# A StackedWAE derivation details

## A.1 Marginal constraint

The space of couplings $\Gamma \in \mathcal{P}(P_D, P_\Theta)$ that defines the OT distance in Eq. (3) is constrained according to Eq. (4). WAE assumes a joint density of the form given in Eq. (5), which automatically satisfies the $p_D$ marginal constraint, but requires the further sufficient condition given in Eq. (6) in order to satisfy the $p_\Theta$ marginal constraint. To see that Eq. (6) is indeed a sufficient condition for the $p_\Theta$ marginal constraint, note that from the Markovian assumption of the generative model (see Eq. (1)), we can write $\gamma$ as
$\forall (x, \tilde{x}) \in \mathcal{X} \times \mathcal{X}$,

$$
\begin{aligned}
\gamma(x, \tilde{x}) &= \int_{\mathcal{Z}_1 \ldots \mathcal{Z}_N} p_{\theta_1}(\tilde{x}|z_1) \, q_{\phi_1}(z_1, \ldots, z_N|x) \, p_D(x) \, dz_1 \ldots dz_N \\
&= \int_{\mathcal{Z}_1} p_{\theta_1}(\tilde{x}|z_1) \, q_{\phi_1}(z_1|x) \, p_D(x) \, dz_1
\end{aligned}
\tag{13}
$$

The constraint in Eq. (4) on the $p_D$ marginal is trivially true as the integral over the second variable can be brought inside the integral over $\mathcal{Z}_1$, after which all the integrals simply integrate to unity leaving $p_D$.

The constraint on the second marginal is obtained by integrating Eq. (13) over the first variable,
$\forall x \in \mathcal{X}$,

$$
\begin{aligned}
\int_{\mathcal{X}} \gamma(x, \tilde{x}) \, dx &= \int_{\mathcal{X}} \int_{\mathcal{Z}_1} p_{\theta_1}(\tilde{x}|z_1) q_{\phi_1}(z_1|x) p_D(x) dz_1 \, dx \\
&= \int_{\mathcal{Z}_1} p_{\theta_1}(\tilde{x}|z_1) \int_{\mathcal{X}} q_{\phi_1}(z_1|x) p_D(x) dx \, dz_1
\end{aligned}
\tag{14}
$$

Thus, to satisfy Eq. (4), we need:
$\forall \tilde{x} \in \mathcal{X}$,

$$
\int_{\mathcal{Z}_1} p_\Theta(\tilde{x}|z_1) \underbrace{\int_{\mathcal{X}} q_{\phi_1}(z_1|x) p_D(x) dx}_{\overset{\text{def}}{=} q_1^{\text{agg}}(z_1)} \, dz_1 \overset{\text{need}}{=} p_\Theta(\tilde{x})
\tag{15}
$$

$$
\overset{\text{def}}{=} \int_{\mathcal{Z}_1} p_{\theta_1}(\tilde{x}|z_1) \, p_{\Theta_{2:N}}(z_1) \, dz_1
$$

where the definition of the generative model from Eq. (1) was used and we introduced:
$\forall z_1 \in \mathcal{Z}_1$,

$$
p_{\Theta_{2:N}}(z_1) \overset{\text{def}}{=} \int_{\mathcal{Z}_2 \ldots \mathcal{Z}_N} p_{\theta_2}(z_1|z_2) \ldots p_{\theta_N}(z_{N-1}|z_N) \, p(z_N) \, dz_2 \ldots dz_N
\tag{16}
$$

To satisfy Eq. (15), one obvious sufficient condition on the aggregated posterior distribution $Q_1^{\text{agg}}(Z_1)$ defined in Eq. (8) is Eq. (6), namely that

$$
\forall z_1 \in \mathcal{Z}_1, \quad q_1^{\text{agg}}(z_1) = p_{\Theta_{2:N}}(z_1)
\tag{17}
$$

which is what we sought out to show. However, Eq. (17) is a sufficient condition, not a necessary one: indeed Eq. (17) must only hold under $\int_{\mathcal{Z}_1} P_\Theta(\tilde{X}|z_1) dz_1$. So for example, if $P_{\theta_1}(\tilde{X}|z_1) = P_{\theta_1}(\tilde{X})$, then Eq. (15) would boil down to a constraint only on the expectations of $Q_1^{\text{agg}}(Z_1)$ and $P_{\Theta_{2:N}}(Z_1)$.

## A.2 WAE OBJECTIVE

Starting from the definition of the OT distance given in Eq. (3), and using the WAE approach with density $\gamma$ written as Eq. (13), we find:

$$\text{OT}_c(P_D, P_\Theta) = \inf_{\Gamma \in \mathcal{P}(P_D, P_\Theta)} \int_{\mathcal{X} \times \mathcal{X}} c(x, \tilde{x}) \, d\Gamma(x, \tilde{x}) \tag{18}$$

$$\leq \inf_{\substack{Q_{\phi_1}(Z_1, Z_2, \ldots Z_n | X), \\ \int_{\mathcal{Z}_1} p_\Theta(\tilde{x}|z_1) q_1^{\text{agg}}(z_1) dz_1 = \int_{\mathcal{Z}_1} p_\Theta(\tilde{x}|z_1) p_{\Theta_{2:N}}(z_1) dz_1}} \int_{\mathcal{X} \times \mathcal{X}} c(x, \tilde{x}) \int_{\mathcal{Z}_1} p_{\theta_1}(\tilde{x}|z_1) q_{\phi_1}(z_1 | x) p_D(x) dz_1 \, dx d\tilde{x}$$

Given that the above does not depend on $z_{>1}$, the inf can be written over $Q_{\phi_1}(Z_1 | X)$ rather than the full $Q_{\phi_1}(Z_1, Z_2, \ldots Z_n | X)$. Replacing the constraint in the inf with the sufficient condition according to Eq. (6), which amounts to replacing Eq. (15) with Eq. (17), we obtain:

$$\text{OT}_c(P_D, P_\Theta) \leq \inf_{\substack{Q_{\phi_1}(Z_1 | X), \\ Q_1^{\text{agg}} = P_{\Theta_{2:N}}}} \int_{\mathcal{X} \times \mathcal{X}} c(x, \tilde{x}) \int_{\mathcal{Z}_1} p_{\theta_1}(\tilde{x}|z_1) q_{\phi_1}(z_1 | x) p_D(x) dz_1 \, dx d\tilde{x} \tag{19}$$

Eq. (7) is then obtained by relaxing constraint in Eq. (19); replacing the hard constraint by a soft constraint via a penalty term added to the objective, weighted by a $\lambda_1$:

$$\widehat{W}_c(P_D, P_\Theta) = \inf_{Q_{\phi_1}(z_1 | x)} \int_{\mathcal{X} \times \mathcal{X}} \int_{\mathcal{Z}_1} c(x, \tilde{x}) \, p_{\theta_1}(\tilde{x}|z_1) \, q_{\phi_1}(z_1 | x) \, p_D(x) \, dz_1 dx d\tilde{x} \tag{20}$$
$$+ \lambda_1 \, \mathcal{D}_1 \Big( Q_1^{\text{agg}}(Z_1), \, P_{\Theta_{2:N}}(Z_1) \Big)$$

where $\mathcal{D}_1$ is any divergence function between distributions on $\mathcal{Z}_1$.

# B EXPERIMENTS

## B.1 MNIST EXPERIMENTS

We train a deep-hierarchical latent-variable model with $N = 5$ latent layers whose dimensions are $d_{\mathcal{Z}_1} = 32$, $d_{\mathcal{Z}_2} = 16$, $d_{\mathcal{Z}_3} = 8$, $d_{\mathcal{Z}_4} = 4$ and $d_{\mathcal{Z}_5} = 2$, respectively. We parametrise the generative and inference models as:

$$\begin{aligned}
q_{\phi_i}(z_i | z_{i-1}) &= \mathcal{N}\big(z_i; \mu_i^q(z_{i-1}), \Sigma_i^q(z_{i-1})\big), \quad i = 1, \ldots, 5 \\
p_{\theta_i}(z_{i-1} | z_i) &= \mathcal{N}\big(z_{i-1}; \mu_i^p(z_i), \Sigma_i^p(z_i)\big), \quad i = 2, \ldots, 5 \\
p_{\theta_1}(x | z_1) &= \delta\big(x - f_{\theta_1}(z_1)\big)
\end{aligned} \tag{21}$$

For both the encoder and decoder, the mean and diagonal covariance functions $\mu_i, \Sigma_i$ are fully-connected networks with 2 same-size hidden layers (consider $f_{\theta_1}$ as $\mu_1^p$). For $i = 1, 2, 3, 4, 5$, the number of units is 2048, 1024, 512, 256, 128, respectively.

For the regularisation hyperparameters, we use $\prod_{i=1}^n \lambda_i = \lambda_{\text{rec}}^n / d_{z_n}$ for $n = 1, \ldots, 4$ (for each reconstruction term in the objective), and $\prod_{i=1}^5 \lambda_i = \lambda_{\text{match}}$ (for the final divergence term). We then perform a grid search over the 25 pairs $(\lambda_{\text{rec}}, \lambda_{\text{match}}) \in \{0.01, 0.05, 0.1, 0.5, 1\} \otimes \{10^{-4}, 10^{-3}, 10^{-2}, 10^{-1}, 1\}$ and find the best result (smallest Eq. (10)) is obtained with $(\lambda_{\text{rec}}, \lambda_{\text{match}}) = (0.05, 10^{-4})$.

We choose the squared Euclidean distance as the cost function: $c_n(z_n, \tilde{z}_n) = \|z_n - \tilde{z}_n\|_{L^2}^2$. The expectations in Eq. (10) are computed analytically whenever possible, and with Monte Carlo sampling otherwise.

We use batch normalisation (Ioffe & Szegedy, 2015) after each hidden fully-connected layer, followed by a ReLU activation (Glorot et al., 2011). We train the models over $5,000$ epochs using Adam optimiser (Kingma & Ba, 2015) with default parameters and batch size of 128.

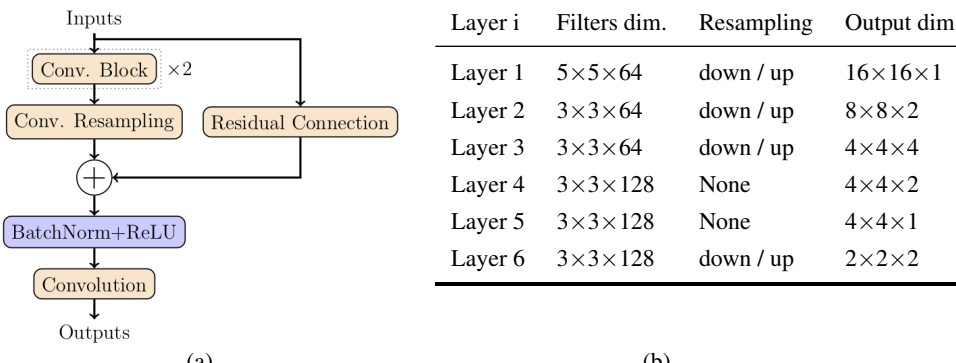

| Layer i | Filters dim. | Resampling | Output dim. |
|---------|-------------|------------|-------------|
| Layer 1 | 5×5×64 | down / up | 16×16×1 |
| Layer 2 | 3×3×64 | down / up | 8×8×2 |
| Layer 3 | 3×3×64 | down / up | 4×4×4 |
| Layer 4 | 3×3×128 | None | 4×4×2 |
| Layer 5 | 3×3×128 | None | 4×4×1 |
| Layer 6 | 3×3×128 | down / up | 2×2×2 |

(a)                                                                  (b)

Figure 6: (a) Residual network with 3 hidden convolutions. (b) Details of the architecture use in Section 3.2.

### B.2 SVHN EXPERIMENT

SVHN SETUP

We train a 6-layers STACKEDWAE on the SVHN dataset using both the training dataset (73,257 digits) and the additional training dataset (531,131 digits). The mean and covariance functions of the inference networks and generative models are parametrised by 3-layer ResNet-style (Kaiming et al., 2015) neural networks.

A $M$-layer residual network is composed of $M - 1$ convolutional blocks followed by a resampling convolution, and a residual connection. The outputs of the two are added and a last operation (either fully connected or convolution layer) is applied on the result. A convolutional block is composed of a convolution layer followed by batch normalisation (Ioffe & Szegedy, 2015) and a ReLU non-linearity (Glorot et al., 2011). We also use batch normalisation and ReLU after the sum of the convolutional blocks output and the residual connection. See Figure 6a for an example of a 3-layers residual network with a last convolution operation.

When resampling, we use a convolution layer with stride 2 in both the skip connection and the resampling covolution for the inference networks and a deconvolution layer with stride 2 in both the skip connection and the resampling convolution in the generative models. If no resampling is performed, then the resampling convolution is a simple convolution layer with stride 1 and the skip connection performs the identity operation. The latent dimensions are then given by the dimensions and the number of features in the last convolutional operation. More specifically, we use $M = 2$ convolutional blocks with the dimensions of the filters specified in Table 6b. Networks in layers $1, 2, 3$ have 96 convolution filters while whose in layers $4, 5, 6$ have 128 filters, each filters having the same size within each residual network, and doubling (in the inference networks) or divide by 2 (in the generative models) their number in the resampling convolution if any resampling is performed. The latent layers $1, 2, 3$ and $6$ have a stride of 2 and we choose the number of features to be $1, 2, 4, 2, 1, 1$ for the latent layers $i = 1 \ldots 6$ (see Table 6b for the full details).

For the regularisation hyperparameters, we use $\prod_{i=1}^{p} \lambda_i = \lambda_{\text{rec}}^{(p-1)/2+1}$ for $p = 1, \ldots, 5$ (each reconstruction term in the objective), and $\prod_{i=1}^{6} \lambda_i = \lambda_{\text{match}}$ (the final divergence term). The choice for the reconstruction weights is motivated by the fact that the effective regularisation hyperparameters scale exponentially. Thus, to avoid the collapse (or blowing up for $\lambda_{\text{rec}} > 1$) of the corresponding reconstruction terms, we choose the weights to scale like $\mathcal{O}\left(\lambda_{\text{rec}}^{p/2}\right)$. We found that $\left(\lambda_{\text{rec}}, \lambda_{\text{match}}\right) = \left(10^{-1}, 10^{-4}\right)$ worked well with our experimental setup.

We train the models over 1000 epochs using Adam optimiser (Kingma & Ba, 2015) with default parameters and batch size of 100.

