# OpenReview forum: "Learning Deep-Latent Hierarchies by Stacking Wasserstein Autoencoders"
_ICLR.cc/2020/Conference — Reject_

### Official Review · AnonReviewer1 · 2019-10-18
**Official Blind Review #1**

**Rating:** 1

**Review:**

The paper aims to develop a deep generative model, which -unlike VAEs or GANs- comprises a hierarchy of latent variables rather than a direct map from the stochastic latent manifold to the observation space. To this end, the paper builds a training objective based on nesting the Wasserstein distance between the data distribution and its estimation arbitrarily many times. The generated objective corresponds naturally to a deep hierarchical generative model.

The principled approach followed to achieve the objective is solid and elegant. It is also intuitive and matches nicely with some valid observations highlighted in the paper such as insufficiency of by-passing intermediate latent variables (sentence above the Sec 2.3 title).

One major weakness of the paper is that it lacks a sufficient argumentation about how it differentiates from earlier attempts to nest Wasserstein distances. For instance,

Y. Dukler et al., "Wasserstein of Wasserstein Loss for Learning Generative Models", ICML, 2019

Apart from the theoretical argumentation, the paper should also compare their solution to this prior work on a number of benchmarks.

Another major weakness is that the paper lacks a quantitative evaluation scheme for its success. The experiments section starts with the claim that the proposed method "significantly" improves on the WAE, which I fail to see on the plots.

Lastly, Having said that the proposed method is novel and elegant, it is still a straightforward extension of the existing and well-known Wasserstein Auto-Encoder (WAE) approach. It extends WAEs by repetitively applying the tricks proposed by this earlier work, putting aside some minor additional adjustments.

Minor on style: The abstract does not give any single hint about the methodological novelty of the work.

---
Post-rebuttal: Thanks to authors for their effort for clarifications. Yet, I'm afraid the author response does not touch at all to any of the concerns I have raised. There are well-known ways to compare the success of generative models, FID being one of them as the authors point out. Another could be the test log-likelihood of a synthetic data set the true distribution of which can be predesigned. I understand the issues the authors raise about the difficulties in comparing generative models, but I kindly disagree with the attitude that there are no ways to compare, so we are obliged to live with qualitative comparisons. If a one-score comparison is not enough, the right way to go is to provide multiple scores. If direct metrics are not feasible, one should go for indirect ones, but should still provide outcomes a reader can reproduce.

**Experience Assessment:**

I have read many papers in this area.

**Review Assessment: Checking Correctness Of Derivations And Theory:**

I did not assess the derivations or theory.

**Review Assessment: Checking Correctness Of Experiments:**

I assessed the sensibility of the experiments.

**Review Assessment: Thoroughness In Paper Reading:**

I made a quick assessment of this paper.

---

> ### Author Response · Authors · 2019-11-11
> **On the comparison with prior works.**
>
> We thank the reviewer for their feedback.
>
> The reviewer identified 3 areas on which they felt the draft could be stronger: (i) incrementality versus[1], (ii) quantitative evaluations of Stacked WAE versus other methods, and (iii) purported lack of significance in addition to the standard WAE framework. We believe that, unfortunately, this review has misunderstood our work. In particular:
>
> (i) the comparison with [1] misunderstands the difference between our approach to "stacking" WAE losses and the "nesting" of Wasserstein distances in [1],
>
> (ii) the lack of quantitative comparisons is a result of our work using a novel non-likelihood based objective, which prohibits natural single-metric comparisons, and
>
> (iii) our work is a mathematically clean and qualitatively incremental contribution on top of the existing WAE approach, enabling the training of latent-variable models that the WAE outright fails to train.
>
> We address each of these points in turn.
>
> [1] propose to nest Wasserstein distances in the sense that they use the Wasserstein distance in the space of the images pixels as their $\textit{ground metric}$. They then use the dual formulation of the 1-Wasserstein distance (in the image space) to derive an adversarial objective for training generative models. This differs from our work in 2 paradigmatic ways. Firstly, while [1] use the "nested" Wasserstein distance as their ground metric for the Wasserstein distance, we use the "nested" Wasserstein distance as a regularisation term on the space of latent distributions in the formulation of the WAE objective. Secondly, the objective in[1] is trained using an adversarial scheme and thus, no encoder network allows for the mapping from the observation space to the latent space. In our work, we are interested in training deep-hierarchical generative models in the autoencoder framework, with an encoder network allowing us to learn a meaningful latent manifold. The role of the "nested" Wasserstein distance in these two works is thus only the same in nomenclature: [1] actually $\textit{nest}$ a Wasserstein distance in the pixel space as their ground metric, while we $\textit{stack}$ a Wasserstein distance as a latent regulariser in the WAE objective.
>
> We agree with the reviewer that a rigorous comparison with existing methods is important. That said, the form of the WAE loss makes such a comparison hard. Indeed, in the WAE, the relaxation of the hard constraint on the coupling of the data distribution and the generative distribution introduces a hyperparameter that will be tuned for each experiments. Moreover, the WAE objective is a likelihood free method, making it hard to compare with the common likelihood based methods. A good metric that enables the comparisons between likelihood and non-likelihood methods remains to be discovered. One attempt at comparing generative models trained with non-comparable objectives is to use sample-based metrics such as the FID score ([2]). However, given the data sets considered in our work, we felt that such metric would not be relevant. Despite this, we do perform a qualitative comparison with the original WAE method when training deep hierarchical models. We intuitively explain why WAEs would fail in training deep hierarchical latent models in section 2.2 (see Equation (7)) and then show empirically in section 3.1 that it is indeed the case (see Figure 4). While the 5-layer generative model trained as WAE achieved good reconstructions (Figure 4a), the samples are significantly worse than those obtained using our Stacked WAE (Figure 4b versus Figure 2b) and no structure was learnt in the deep latent space (Figure 4c versus Figure 2c).
>
> Finally, while our Stacked WAE method is indeed built on the well-known WAE objective and consists of stacking WAE modules on the top of each other, the novelty resides in the way we unroll the original WAE objective, using WAEs as latent regularisers at each layer, enabling the hierarchical model to leverage all of its deep layers. This allows for the propagation of information from the observation space all the way to the deepest latent layer in fully factorised Markov models, and by doing so, it captures the data structure all along the hierarchy. This result, which we clearly demonstrate, is something that both WAEs and VAEs outright fail at. In this sense we do not consider our work to be an insignificant contribution on top of the pre-existing WAE framework.
>
> We hope that this review might either be amended significantly given that it seems to have misunderstood both our work and the relevant literature.
>
> [1]: Y. Dukler,  W. Li, A. Lin and G. Montufar. Wasserstein of Wasserstein Loss for Learning Generative Models. In International Conference on Machine Learning, 2019.
> [2]: M. Heusel, H. Ramsauer, T. Unterthiner, B. Nessler, and S. Hochreiter. Gans trained by a two time-scale update rule converge to a local nash equilibrium. In Advances in neural information processing systems, 2017.

---

### Official Review · AnonReviewer2 · 2019-10-24
**Official Blind Review #2**

**Rating:** 3

**Review:**

In this paper, a hierarchical extension to Wasserstein Autoencoders (WAE) is proposed, where the latent variables are stacked in a multi-layer structure. In the proposed model, the divergence function in WAE is viewed as a relaxed WS distance. Therefore, another layer of WAE can be stacked to minimise the WS distance. In this way, a hierarchical model can be built to learn hierarchical representations.

I think the idea of viewing the divergence in WAE as a relaxed WS distance and then minimising it with another WAE structure is interesting, intuitive and straightforward. However, the advantages of the proposed model over WAE and VLAE (S.Zhao et.al 2017) are less obvious to me. It is a bit hard for me to tell whether the hierarchical latent variables help to improve quantitative results, generate better images, or learn intuitive hierarchical representations, which is the main reason that I go to mild rejection.

For example, I would expect to see similar things as in VLAE, where the representations in different layers capture hierarchical structures or disentanglements. But in the proposed model, it seems to be hard to see the differences between the hierarchical representations such as in Figure 3(b). Also in the two-dimensional visualisation of Figure 3(a), it is a bit hard for me to intuitively understand what the representations really capture.

From the graphical model point of view, the proposed model is a hierarchical Gaussian model and the inference (although with WAE) is in the flavour of Gibbs sampling, which propagates information layer-wisely from bottom up. Conventionally, a hierarchical Gaussian model is hard to work with many layers such as 5. Therefore, I may suggest improving in case of fewer layers.

**Experience Assessment:**

I have read many papers in this area.

**Review Assessment: Checking Correctness Of Derivations And Theory:**

I assessed the sensibility of the derivations and theory.

**Review Assessment: Checking Correctness Of Experiments:**

I assessed the sensibility of the experiments.

**Review Assessment: Thoroughness In Paper Reading:**

I read the paper at least twice and used my best judgement in assessing the paper.

---

> ### Author Response · Authors · 2019-11-11
> **On the motivation behind training deep latent hierarchical generative models.**
>
> We appreciate the reviewer detailed feedback.
>
> There are many differences between our work and that of [1], but they stem predominantly from a significant difference in motivation. In [1], the authors train a single-latent-layer generative model (in evident contrast to our work) with a bespoke architecture for the encoder and decoder aiming at capturing hierarchical structure in the data and learning disentangled representations. In our work on the other hand, the goal was to show that using the Stacked WAE objective, a deep-hierarchical-latent model can be trained, in principle improving generative capacity over shallower generative models.
>
> While we acknowledge that one of the motivations behind using hierarchical latent-variable models is the discovery of hierarchical representations (as [1] sought to do), we focus on improving the ability of generative models to learn deep latent hierarchies (similarly to [2], [3]). That is, our motivation is to methodologically enable the training of deep latent hierarchies. Indeed, as explained in Section 2.2 and shown in Section 3.1, the Stacked WAE method allows for better training of deep hierarchical generative models than the original WAE framework. More specifically, it is able to learn an approximate posterior over all the latent layers as opposed to the WAE, and without the need for skip connections and weight sharing in the encoder and decoder networks unlike VAE methods ([2], [3]).
>
> We admit that this leads to debatable choices for the generative models considered. For example, in the MNIST experiment in Section 3.1, we trained a generative model that is surely too deep for this simple data set (we use 5 latent layers while [1] have only 3 levels in their hierarchy). The intention of our work was not to learn MNIST well, but to show that a 5-layer latent-variable model can actually be trained on MNIST (a feat that requires significant architecture and optimisation design in the VAE setting, see Figure 6 of [2]). This is why we did not, for example, carefully interpret our latent hierarchies, despite that being an interesting question.
>
> To make our motivation clearer to readers, in particular in contrast to [1], we have added a short discussion to the introduction. We believe that the motivational differences between our work and that of [1] justify the shortcomings pointed out by the reviewer, and in this context hope that the reviewer would agree to amending their rating to a "weak accept".
>
> [1]: S. Zhao, J. Song, and S. Ermon.  Learning hierarchical features from deep generative models. In International Conference on Machine Learning, 2017.
> [2]: C. K. Sønderby, T. Raiko, L. Maaløe, S. K. Sønderby, and O. Winther. Ladder variational autoencoders. In Advances in neural information processing systems, 2016.
> [3]: L. Maaløe, M. Fraccaro, V. Liévin, and O. Winther. BIVA: a very deep hierarchy of latent variables for generative modeling. In Advances in neural information processing systems, 2019.

---

### Official Review · AnonReviewer3 · 2019-10-29
**Official Blind Review #3**

**Rating:** 6

**Review:**

This paper presents a deep, latent variable model for unsupervised data modeling problems. The problem with such latent, deep generative models is that they are difficult to train reliably. In this paper, the authors provide an approach based on stacked Wasserstein autoencoders to train deep latent variable models. Experimental results are demonstrated on various image datasets and the latent codes are demonstrated to have an interpretable meaning.
I like the inference techniques in the paper and like the ideas presented in this paper.

**Experience Assessment:**

I do not know much about this area.

**Review Assessment: Checking Correctness Of Derivations And Theory:**

N/A

**Review Assessment: Checking Correctness Of Experiments:**

I assessed the sensibility of the experiments.

**Review Assessment: Thoroughness In Paper Reading:**

I made a quick assessment of this paper.

---

> ### Author Response · Authors · 2019-11-11
> **Thank you**
>
> We thank the reviewer for their positive feedback.

---

### Decision · Program_Chairs · 2019-12-19

**Decision:**

Reject

**Comment:**

The paper received 6, 3, 1. The main criticism is the lack of quantitative evaluation/comparison. The rebuttal did not convince the last reviewer who strongly argues for a comparison. The authors are encouraged to add additional results and resubmit to a future venue.